# Effectiveness of first-line treatment for relapsing-remitting multiple sclerosis in Brazil: A 16-year non-concurrent cohort study

Kathiaja Miranda Souza[1]*, Isabela Maia Diniz[2,3], Lívia Lovato Pires de Lemos[2,4], Nélio Gomes Ribeiro Junior[2], Isabella de Figueiredo Zuppo[2], Juliana Alvares Teodoro[2,3], Francisco de Assis Acurcio[2,3,4], Álvaro Nagib Atallah[1,5]*, Augusto Afonso Guerra Júnior[2,3]

1 Programa de Pós-Graduação em Saúde Baseada em Evidências, Universidade Federal de São Paulo, São Paulo, São Paulo, Brazil, 2 SUS Collaborating Centre for Technology Assessment & Excellence in Health, Faculdade de Farmácia, Universidade Federal de Minas Gerais, Belo Horizonte, Minas Gerais, Brazil, 3 Programa de Pós-Graduação em Medicamentos e Assistência Farmacêutica, Faculdade de Farmácia, Universidade Federal de Minas Gerais, Belo Horizonte, Minas Gerais, Brazil, 4 Programa de Pós-Graduação em Saúde Pública, Faculdade de Medicina, Universidade Federal de Minas Gerais, Belo Horizonte, Minas Gerais, Brazil, 5 Cochrane Brazil, Centro de Estudos de Saúde Baseada em Evidências e Avaliação Tecnológica em Saúde, São Paulo, São Paulo, Brazil

* kathiajamsouza@gmail.com (KMS); atallahmbe@uol.com.br (ANA)

**Data Availability Statement:** There are ethical and legal restrictions on sharing the data set used in this study. The SUS databases are not open for

## Abstract

### Background

Relapsing-remitting multiple sclerosis (RRMM) is a chronic, progressive, inflammatory and immune-mediated disease that affects the central nervous system and is characterized by episodes of neurological dysfunction followed by a period of remission. The pharmacological strategy aims to delay the progression of the disease and prevent relapse. Interferon beta and glatiramer are commonly used in the Brazilian public health system and are available to patients who meet the guideline criteria. The scenario of multiple treatments available and in development brings the need for discussion and evaluation of the technologies already available before the incorporation of new drugs. This study analyses the effectiveness of first-line treatment of RRMS measured by real-world evidence data, from the Brazilian National Health System (SUS).

### Methods and findings

We conducted a non-concurrent national cohort between 2000 and 2015. The study population consisted of 22,722 patients with RRMS using one of the following first-line drugs of interest: glatiramer or one of three presentations of interferon beta. Kaplan–Meier analysis was used to estimate the time to treatment failure. A univariate and multivariate Cox proportional hazard model was used to evaluate factors associated with treatment failure. In addition, patients were propensity score-matched (1:1) in six groups of comparative first-line treatments to evaluate the effectiveness among them. The analysis indicated a higher risk of treatment failure in female patients (HR = 1.08; P = 0,01), those with comorbidities at baseline (HR = 1.20; P<0,0001), in patients who developed comorbidities after starting

public review with strict regulations, federal laws and requirements for access. The data also contains sensitive patient information like physical addresses, location data, and specific dates, such as birth dates, death dates, and procedures dates. According to the opinion by the Ethics Committee of the University Hospital at the Federal University of São Paulo, the data set must be kept confidential. However, other researchers may request further information and data access from Professor Dr. Mariângela Leal Cherchiglia of the Department of Preventive and Social Medicine, College of Medicine, Federal University of Minas Gerais, Brazil and SUS Collaborating Centre—Technology Assessment & Excellence in Health, College of Pharmacy, Federal University of Minas Gerais, Brazil (ccates@ccates.org.br; cherchml@medicina.ufmg.br).

**Funding:** This work was supported by the SUS Collaborating Centre Technology Assessment & Excellence in Health and of the Federal University of São Paulo – UNIFESP. The funders had no role in study design, data collection and analysis, decision to publish, or preparation of the manuscript.

**Competing interests:** The authors have declared that no competing interests exist.

treatment (i.e., rheumatoid arthritis—HR = 1.65; P<0,0001), those exclusive SUS patients (HR = 1.31; P<0,0001) and among patients using intramuscular interferon beta (IM βINF-1a) (28% to 60% compared to the other three treatments; P<0,0001). Lower risk of treatment failure was found among patients treated with glatiramer.

## Conclusions

This retrospective cohort suggests that glatiramer is associated with greater effectiveness compared to the three presentations of interferon beta. When evaluating beta interferons, the results suggest that the intramuscular presentation is not effective in the treatment of multiple sclerosis.

## Introduction

Multiple sclerosis (MS) is a chronic, progressive, inflammatory and immune-mediated disease that affects the central nervous system and is characterized by loss of motor and sensory function resulting from immune-mediated inflammation, demyelination, and subsequent axonal damage. It is characterized by episodes of neurological dysfunction that can be followed by a period of remission or may progress [1].

Relapsing-remitting multiple sclerosis (RRMS) is the most common form of the disease and is identified by relapses or acute attacks (worsening of symptoms of neurological dysfunction lasting 24 hours or longer, in the absence of fever), which can enter remission spontaneously or upon treatment with corticosteroids (pulse therapy). The most commonly observed symptoms are optic neuritis; paresis or paresthesia of limbs; motor incoordination; lack of balance; myelitis; sphincter dysfunction and cognitive-behavioral disorders, either alone or in combination [1–3].

The prevalence rate of MS is variable worldwide, with a global average prevalence of 33 cases per 100,000 inhabitants. In Brazil, a recent systematic review of prevalence studies estimated an average rate of 8.69 cases per 100,000 inhabitants, with considerable variation among the regions of the country, with a prevalence of 1.36 cases per 100,000 inhabitants in the northeast region and 27. 2 cases per 100,000 inhabitants in the south region. The Brazilian Multiple Sclerosis Association (ABEM) estimates that there are 35,000 Brazilians currently living with the disease [4–6].

The treatment, based on the use of Disease Modifying Therapies (DMTs) or immunomodulators, aims to delay the progression of RRMS and prevents relapse by decreasing circulating immune cells or by preventing these cells from crossing the blood-brain barrier, thereby reducing the inflammatory response [7, 8]. Until December 2015, interferon beta (βINF) and glatiramer were the only alternatives for first-line treatment of patients who met the guideline criteria in the Brazilian public health system (SUS). According to international guidelines and the Clinical Guideline of the Brazilian Ministry of Health, patients could start with either one of the first-line treatments and switch to the other in the case of treatment failure. Treatment failure is defined as a presence of relapses, activity on magnetic resonance imaging, a progression of disability or intolerance due to adverse events [9].

The results of primary clinical studies are important sources of information for the listing and maintenance of medicines towards the treatment of RRMS in the health system. However, most studies have limitations such as the use of placebo as a control group, short follow-up protocols, specific eligibility criteria of patients, high costs, little relevant clinical outcomes and

existence of competing interests [10, 11]. These limitations raise the question of whether health systems achieve the expected results when they decide to incorporate and pay for health technologies. Also, it is known that there are more than 12 different treatments commercially available, as well as new medicines that appear in the technological horizon to treat MS [12]. This scenario has a strong impact on SUS since there is an increasing demand for the incorporation of new technologies to treat this condition [13]. This fact brings forth the need for discussion and evaluation of the technologies already available in SUS for a better understanding of the use of these medicines and obtaining adequate information to guide decision-making for SUS sustainability and efficiency [14].

In Brazil, the information on the dispensing of DMT and outpatient procedures are recorded in the SUS Outpatient Information System. Hospital procedures, such as treatment for relapses and complications are recorded in the SUS Hospital Information System. The integration of these databases with the Mortality Information System allows the construction of a robust patient-centered registry for long term longitudinal follow-up of these patients [15, 16]. We believe it is important to analyze the profile of utilization and the effectiveness of first-line DMT in patients with RRMS in the Brazilian public health to guide stakeholders in future investment and disinvestment decisions. We also hope our results will be useful to aid more researches on multiple sclerosis.

## Methods

### Ethical statement

The Research Ethics Committee for the University Hospital at the Federal University of São Paulo (HSP/UNIFESP, processes number 2.468.332 and CAAE 81043417.6.0000.5505) approved of the study, under Brazilian national legislation.

### Study design and setting

This study is a nationwide non-concurrent open cohort with follow-up from 2000 to 2015 developed by deterministic-probabilistic linkage using the patient-centered registry of the Hospital Information System (SIH), Ambulatory Information System (SIA) and Mortality Information System (SIM) [15]. SIH comprises data on hospitalization from both public and private hospitals contracted by SUS. The subsystem of High-Complexity Procedure Authorization (APAC) of the SIA database comprises all information about DMT drugs dispensing and also records the medical diagnosis for which the drug is prescribed, using the International Classification of Diseases, Tenth Revision (ICD-10) codes.

Although in Brazil there is a mix of public (SUS) and private health services, the SUS covers health services for approximately 80% of citizens. Moreover, it is believed most citizens of Brazil use the SUS for high-cost medicines such as the DMT agents [17, 18]. Thus, we believe that SUS database comprehensively includes people with RRMS in Brazil. Previous studies have used the same methodology and proven accuracy of the registers [15, 18, 19].

### Study population

The study population consisted of RRMS patients aged 18 years and older identified by the registry of ICD-10 code G35 and received one of the following first-line drugs of interest: Glatiramer, intramuscular interferon beta 1a (IM βINF- 1a), subcutaneous interferon beta 1a (SC βINF- 1a) and subcutaneous interferon beta 1b (SC βINF- 1b). We considered 4 exposure groups according to the initial drug dispensed at cohort entry (intention to treat). We excluded patients who started the cohort using other DMTs or were using a combination of more than

one drug. Also, as an inclusion criterion, patients were required to have at least 6 months of records based on the dispensation of the drugs of interest to be considered as a truly RRMS case. The date of entry into the cohort corresponded to the date of the first dispensation registry of DMT after continuous DMT use during these 6 months. The entry period was between January 2000 to December 2014, and patients were followed up from January 2000 to December 2015, 16 years. This strategy assured a minimum follow-up of 12 months. Patients were censored if they abandoned or interrupted their treatment for more than three months or at the end of follow-up (right censoring).

## Study outcomes

The effectiveness outcome was the time until treatment failure, defined as the record of a relapse, a switch to another DMT, or death, whichever occurred first. Relapse was defined by the occurrence of methylprednisolone dispensing and/or pulse steroid therapy and/or hospital relapse treatment. For the event of switch treatment, we allowed a tolerance of three consecutive months without dispensing registry until the patient initiated the other drug. A 3-month gap was selected in an attempt to ensure that patients switching DMTs were not misclassified as having discontinued therapy. It is also consistent with the threshold used in a recently published study examining persistence in patients with MS [20].

## Statistical analysis

Demographic and clinical characteristics were presented as mean ± SD or median and lower and upper quartiles range (Q1–Q3) for the continuous variables and as the number (percentage) of subjects for categorical variables. We used the SUS databases to establish the presence of the comorbidity before and after initial treatment based on the Elixhauser Comorbidity, adapted from Quan et al. [21]. We compared treatment groups using univariate comparisons with Kruskal-Wallis or Chi-squared test.

In addition, the cohort patients were paired (groups βINF-1a IM vs. βINF-1b SC; βINF-1a IM vs. glatiramer; βINF-1a IM vs. βINF-1a SC; βINF-1b SC vs. glatiramer; βINF-1b SC vs. βINF-1a SC; and glatiramer vs. βINF-1a SC) based on the propensity score matched to control possible differences between treatment groups. The propensity score is defined by the probability that the individual will receive a specific treatment, conditioned to the studied covariables. Propensity scores were estimated using logistic regression models considering the following variables: age, sex, presence of comorbidities at the beginning of treatment, region of residence, exclusive SUS patient and period of entry into the cohort (baseline characteristics). The nearest neighbor matching within the caliper method was used, with a maximum fixed difference of 0.2 between the propensity scores to select patients with homogeneous baseline characteristics [22, 23].

Time to treatment failure was analyzed using the Kaplan-Meier survival curves and the log-rank test was used to evaluate the difference among drug treatments. Univariate and multivariate Cox proportional hazards models were used to investigate the impact of baseline characteristics on treatment failure. Additionally, the impact of the appearance of comorbidities after starting DMT treatment was investigated. Variables with $p < 0.20$ in the univariate analysis and variables considered clinically relevant were included in the multivariable model. The proportional hazards assumption was evaluated using statistical tests based on the Schoenfeld residuals [24].

We performed two sensitivity analyses. In the first one, we did not impose a time limit on the off-treatment gap for treatment failure event assignment, that is, patients were not censored for abandonment if they returned to index treatment or switched drugs at any time after

interrupting first treatment. If they did not return to treatment, they were censored for loss to follow-up.

In the second sensitivity analysis we considered patients who use the public health system to access the other offered services which include outpatient and hospital procedures (exclusively SUS patients) (a), and patients who use SUS only to get high-cost drugs (b).

Results were considered significant for p-values < 0.05 (two-sided) and a Bonferroni approximation was employed to adjust for the multiple paired comparisons. The adjusted p-value that indicated significance was 0.008 and was computed as (α/number of comparisons), or (0.05/6) [25, 26]. Data analysis was performed using "R" version 3.4.1 (R Foundation for Statistical Computing).

# Results

## Study population

We identified 27,446 patients with RRMS who had at least one of the DMT treatments in the period 2000 to 2015. Of these, we excluded 95 (0.35%) patients whose data indicated an error during the deterministic-probabilistic record linkage; 1,060 (3.9%) patients for being younger than 18 years old at the start of treatment; 2,571 (9.4%) patients who not fulfilled of the criteria of 6 months of continuous dispensation prior to the start of follow-up and; 998 (3.6%) for having started treatment with other drugs for RRMS (azathioprine, fingolimod or natalizumab) or for using more than one drug, yielding 22,722 patients for analysis. Of the included patients, the majority started treatment with SC βINF-1a (35.6%); followed by IM βINF-1a (24.7%); SC βINF-1b (22.3%) and glatiramer (17.4%) (Fig 1).

## Baseline characteristics

Table 1 provides the baseline demographic and clinical characteristics of MS patients. The median (Q1–Q3) age was 37 (29–46) years and 73.3% were women. Most patients were from

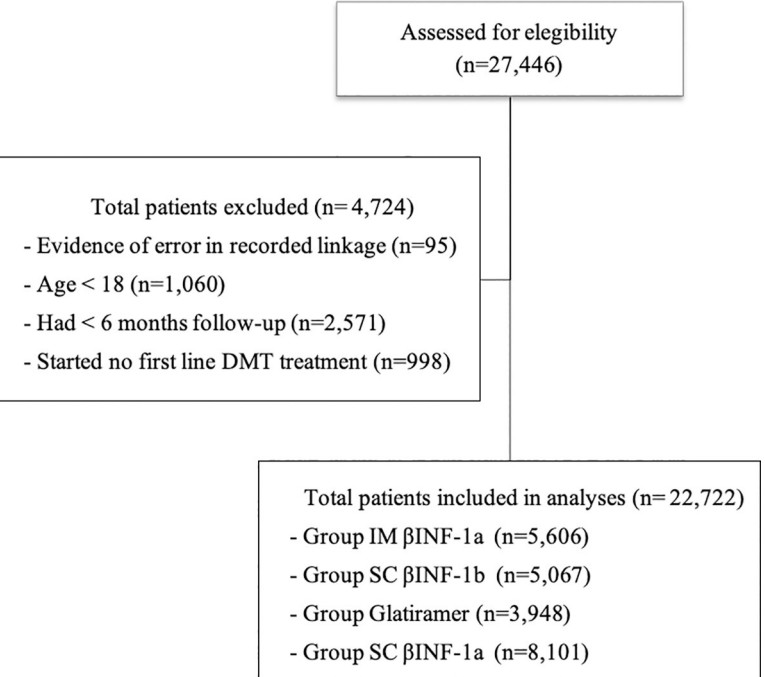

**Fig 1. Flow diagram of the study population.**

**Table 1. Baseline characteristics of Brazilian patients with RRMS included in the study, 2000–2015.**

| | | Initial DMT | | | | |
|---|---|---|---|---|---|---|
| | **Full cohort** | **IM βINF- 1a** | **SC βINF- 1b** | **Glatiramer** | **SC βINF- 1a** | |
| | **n = 22,722** | **n = 5,606** | **n = 5,067** | **n = 3,948** | **n = 8,101** | **P-value** |
| **Gender** | | | | | | <0.0001[a] |
| Female, n (%) | 16,663 (73.3) | 4,163 (74.3) | 3,565 (70.4) | 3,015 (76.4) | 5,920 (73.1) | |
| Male, n (%) | 6,059 (26.7) | 1,443 (25.7) | 1,502 (29.6) | 933 (23.6) | 2,181 (26.9) | |
| **Age** | | | | | | <0.0001[b] |
| Mean (SD) | 37.6 (±11.3) | 37.1 (± 11.2) | 38.2 (± 11.2) | 37.3 (± 11.3) | 38.0 (± 11.4) | |
| Median (Q1-Q3) | 37 (29–46) | 36 (28–45) | 38 (29–46) | 36 (28–45) | 37 (29–46) | |
| **Age group, n (%)** | | | | | | <0.0001[a] |
| 18–25 | 3,438 (15.1) | 880 (15.7) | 710 (14.1) | 619 (15.7) | 1,229 (15.2) | |
| 26–35 | 7,156 (31.5) | 1,858 (33.1) | 1,512 (29.8) | 1,332 (33.7) | 2,454 (30.3) | |
| 36–45 | 6,355 (28.0) | 1,564 (27.9) | 1,483 (29.3) | 1,046 (26.5) | 2,262 (27.9) | |
| 46–55 | 4,241 (18.7) | 934 (16.7) | 1,026 (20.2) | 676 (17.2) | 1,605 (19.8) | |
| 56–65 | 1,303 (5.7) | 317 (5.7) | 285 (5.6) | 242 (6.1) | 459 (5.7) | |
| > 65 | 229 (1.0) | 53 (0.9) | 51 (1.0) | 33 (0.8) | 92 (1.1) | |
| **Period of cohort entry, n (%)** | | | | | | <0.0001[a] |
| 2000–2003 | 4,709 (20.7) | 428 (7.6) | 1,420 (28.1) | 286 (7.2) | 2,575 (31.8) | |
| 2004–2007 | 5,011 (22.1) | 1,362 (24.3) | 1,031 (20.4) | 899 (22.8) | 1,719 (21.2) | |
| 2008–2011 | 8,004 (35.2) | 2,172 (38.8) | 1,847 (36.5) | 1,667 (42.2) | 2,318 (28.6) | |
| 2011–2015 | 4,998 (22.0) | 1,644 (29.3) | 769 (15.0) | 1,096 (27.8) | 1,489 (18.4) | |
| **Region of residence, n (%)** | | | | | | <0.0001[a] |
| North | 354 (1.6) | 52 (0.9) | 93 (1.8) | 34 (0.9) | 175 (2.2) | |
| Northeast | 2,694 (11.9) | 591 (10.5) | 613 (12.1) | 404 (10.2) | 1,086 (13.4) | |
| Midwest | 2,035 (9.0) | 420 (7.5) | 509 (10.0) | 326 (8.2) | 780 (9.6) | |
| Southeast | 13,271 (58.4) | 3,266 (58.2) | 2,987 (59.0) | 2,345 (59.5) | 4,673 (57.7) | |
| South | 4,368 (19.2) | 1,277 (22.8) | 865 (17.1) | 839 (21.2) | 1,387 (17.1) | |
| **Elixhauser comorbidities, n (%)** | | | | | | 0.0045[a] |
| 0 | 20,439 (90.0) | 5,071 (90.4) | 4,509 (89.0) | 3,515 (89.0) | 7,344 (90.6) | |
| 1 | 2,077 (9.1) | 487 (8.7) | 512 (10.1) | 384 (9.8) | 694 (8.6) | |
| ≥2 | 206 (0.9) | 48 (0.9) | 46 (0.9) | 49 (1.2) | 63 (0.8) | |
| **Comorbidity, n (%)** | | | | | | |
| Paralysis | 309 (1.4) | 66 (1.2) | 87 (1.7) | 58 (1.5) | 98 (1.2) | 0.046[a] |
| Rheumatoid arthritis/collagen vascular diseases | 301 (1.3) | 61 (1.1) | 84 (1.6) | 60 (1.5) | 96 (1.2) | 0.028[a] |
| Liver disease | 219 (1.0) | 50 (0.9) | 43 (0.8) | 36 (0.9) | 90 (1.1) | 0.4 [a] |
| Renal failure | 107 (0.5) | 17 (0.3) | 23 (0.5) | 17 (0.4) | 50 (0.6) | 0.065 [a] |
| Psychoses | 100 (0.4) | 21 (0.4) | 24 (0.5) | 29 (0.7) | 26 (0.3) | 0.01 [a] |
| Depression | 59 (0.3) | 11 (0.2) | 9 (0.2) | 19 (0.5) | 20 (0.2) | 0.02 [a] |
| Other neurological disorders | 746 (3.3) | 181 (3.2) | 179 (3.5) | 150 (3.8) | 236 (2.9) | 0.05 [a] |
| Neoplasms | 318 (1.4) | 96 (1.7) | 59 (1.2) | 51 (1.3) | 112 (1.4) | 0.09 [a] |
| Other Comorbidities | 361 (1.6) | 87 (1.5) | 102 (2.0) | 73 (1.8) | 99 (1.2) | 0.002 [a] |
| **Exclusive SUS patients, n (%)** | 9,516 (41.9) | 2,012 (35.9) | 2,468 (48.7) | 1,571 (39.8) | 3,465 (42.8) | <0.0001[a] |

[a] Chi-square tests.

[b] Kruskal-wallis test. **Abbreviations:** IM-IFNβ-1a, intramuscular interferon beta-1a; SC-IFNβ-1b, subcutaneous interferon beta-1b; SC-IFNβ-1a, subcutaneous interferon beta-1a; SUS, Brazilian Public Health System.

the Southeast region (58.4%), followed by South (19.2%), Northeast (11.9%), Midwest (9.0%), and North (1.6%). The majority of patients (35.2%) enrolled in the cohort during the 2008–2011 period and 10% of patients had one or more comorbidities before starting treatment, in which other neurological diseases and paralysis (MS-related comorbidities) were the most frequent.

We observed statistical differences between groups, which were not considered clinically meaningful. Exclusive SUS patients represent 41.9% of all patients, 35.9% of IM βINF-1a group, 39.8% of glatiramer group; 42.8% of SC βINF-1a group; and 48.7% of SC βINF- 1b group.

The characteristics of the matched cohorts are shown in S1 Table. The resulting six paired groups were closely matched on all recorded demographic and clinical parameters. The mean differences between groups in the propensity scores were significantly reduced, with emphasis on the variable period of entry into the cohort, with a reduction of up to 97% of the difference, and up to 62.5% for exclusively SUS patients.

## Treatment failure

After starting initial DMT treatment, the proportion of patients who experienced treatment failure during follow-up was 25.6%, and the IM βINF-1a group (30.3%) had experienced most events compared to other groups (Table 2). The median (SD) follow-up time for this cohort was 68.7 (45.9) months (range 6–192). The median time to treatment failure of all first-line DMT treatment, estimated by Kaplan-Meier survival analysis, was approximately 7 years (88 months, 95% CI 83–96). The median time to treatment failure was worse (i.e., shorter) among female patients; exclusive SUS patients (i.e., users of SUS for other health care procedures); patients that developed comorbidities after starting treatment and among patients of IM βINF-1a group (Fig 2).

The univariate and multivariate analysis indicated that, among all first-line DMT users, higher risk of treatment failure were observed in female patients, those enrolled in the 2000–2003 period, those with comorbidities at baseline, in patients who had comorbidities after starting treatment (MS-related comorbidities, renal failure and rheumatoid arthritis), those exclusive SUS patients and among patients of IM βINF-1a group. Region of residence and other comorbidities were not significant in univariate analysis and were not included in the multivariate analysis (Table 3).

The Kaplan–Meier curves for the matched comparison among the first-line DMT treatment groups are shown in Fig 3. This analysis revealed a significant difference in favor of glatiramer in all comparisons and IM βINF-1a had the worst result in most of the comparisons. Also, SC βINF- 1b presented a shorter time to failure compared to SC βINF- 1a.

**Table 2. Treatment failure among the patients with RRMS included in the study, 2000–2015, Brazil.**

|  | Full cohort | Initial DMT | | | |
|---|---|---|---|---|---|
|  | | IM βINF- 1a | SC βINF- 1b | Glatiramer | SC βINF- 1a |
|  | n = 22,722 | n = 5,606 | n = 5,067 | n = 3,948 | n = 8,101 |
| Treatment failure, n (%) | 5,811 (25.6) | 1,697 (30.3) | 1,371 (27.1) | 806 (20.4) | 1,937 (23.9) |
| Relapses, n (%) | 933 (4.1) | 177 (3.2) | 251 (5.0) | 181 (4.6) | 324 (4.0) |
| Switched the medication, n (%) | 4,839 (21.3) | 1,516 (27.0) | 1,111 (21.9) | 616 (15.6) | 1596 (19.7) |
| Death, n (%) | 39 (0.2) | 4 (0.1) | 9 (0.2) | 9 (0.2) | 17 (0.2) |

**Abbreviations:** DMT, disease-modifying therapy, IM-IFNβ-1a, intramuscular interferon beta-1a; SC-IFNβ-1b, subcutaneous interferon beta-1b; SC-IFNβ-1a, subcutaneous interferon beta-1a.

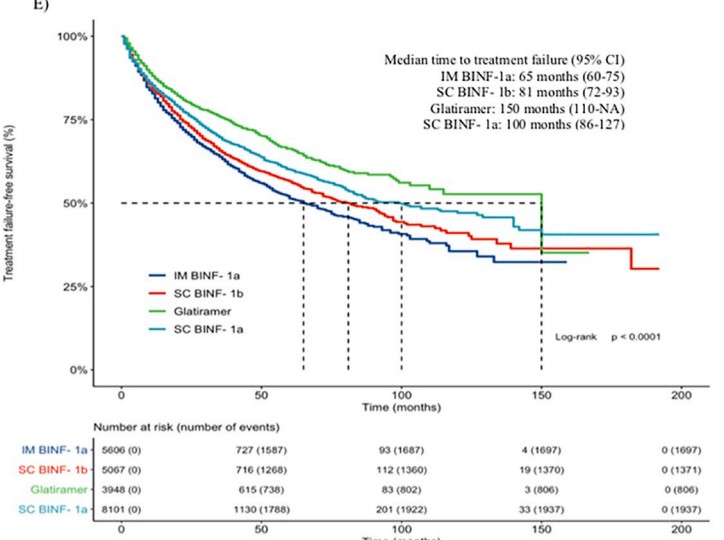

**Fig 2.** Kaplan–Meier estimates for the time to treatment failure of all (A), exclusive and non-exclusive SUS patients (B), gender (C), comorbidities after starting treatment (D) and initial DMT treatment (E). **Abbreviations:** DMT, disease-modifying therapy; CI, confidence interval; IM-IFNβ-1a, intramuscular interferon beta-1a; SC-IFNβ-1b, subcutaneous interferon beta-1b; SC-IFNβ-1a, subcutaneous interferon beta-1a.

The risk of therapeutic failure of IM βINF-1a after pairing by propensity score was 28%; 60% and 41% higher than βINF-1b SC, glatiramer and βINF-1a SC, respectively. We observed that the risk was slightly lower compared to the results of the multivariate analysis. Glatiramer had a lower risk of failure compared to βINF-1b SC and βINF-1a SC by 25% and 13% respectively; however, there was no difference after Bonferroni's adjustment (Table 4).

## Sensitivity analysis

When we did not impose a time limit on the off-treatment gap for the treatment failure event assignment revealed similar results as the main analysis. The median time to treatment failure

**Table 3.  Univariate and multivariate Cox regression models with predictors of DMT treatment failure for RRMS patients included in the cohort, 2000 to 2015, Brazil.**

| Variables* | Univariate | | Multivariate | |
|---|---|---|---|---|
| | HR (95% CI) | P-value | HR (95% CI) | P-value |
| **Gender (Female)** | **1.07 (1.01–1.14)** | **0.022** | **1.08 (1.02–1.14)** | **0.011** |
| **Age group at index date** | | | | |
| 18–25 | -- | | -- | |
| 26–35 | 0.92 (0.85–0.99) | <0.05 | 0.91 (0.84–0.99) | <0.05 |
| 36–45 | 0.89 (0.82–0.97) | <0.01 | 0.84 (0.77–0.91) | <0.0001 |
| 46–55 | 0.82 (0.75–0.90) | <0.0001 | 0.77 (0.71–0.84) | <0.0001 |
| 46–55 | 0.69 (0.60–0.80) | <0.0001 | 0.68 (0.59–0.78) | <0.0001 |
| > 65 | 0.92 (0.70–1.21) | ns | 0.89 (0.67–1.17) | ns |
| **Period of cohort entry** | | | | |
| 2000–2003 | -- | | -- | |
| 2004–2007 | 0.78 (0.73–0.84) | <0.0001 | 0.79 (0.73–0.86) | <0.0001 |
| 2008–2011 | 0.62 (0.57–0.66) | <0.0001 | 0.63 (0.58–0.67) | <0.0001 |
| 2011–2015 | 0.66 (0.62–0.72) | <0.0001 | 0.70 (0.64–0.76) | <0.0001 |
| **Baseline comorbidities** | 1.28 (1.18–1.39) | <0.0001 | 1.20 (1.11–1.31) | <0.0001 |
| **Comorbidities after starting DMT treatment*** | | | | |
| Paralysis | 1.52 (1.37–1.67) | <0.0001 | 1.20 (1.09–1.33) | 0.0007 |
| Rheumatoid arthritis/collagen vascular diseases | 2.13 (1.90–2.4) | <0.0001 | 1.65 (1.47–1.86) | <0.0001 |
| Renal failure | 1.87 (1.61–2.17) | <0.0001 | 1.36 (1.17–1.59) | <0.0001 |
| Other neurological disorders | 1.48 (1.36–1.61) | <0.0001 | 1.18 (1.08–1.29) | 0.0003 |
| **Exclusive SUS patients** | 1.44 (1.36–1.51) | <0.0001 | 1.31 (1.24–1.39) | <0.0001 |
| **Initial DMT** | | | | |
| IM βINF- 1a | -- | | -- | |
| SC βINF- 1b | 0.89 (0.83–0.96) | 0.0017 | 0.78 (0.73–0.84) | <0.0001 |
| Glatiramer | 0.63 (0.58–0.68) | <0.0001 | 0.60 (0.55–0.65) | <0.0001 |
| SC βINF- 1a | 0.80 (0.75–0.85) | <0.0001 | 0.70 (0.66–0.75) | <0.0001 |

* Region of residence and other comorbidities were not significant and not included in the multivariate model.

**Abbreviations**: RRMS, relapsing-remitting multiple sclerosis; DMT, disease-modifying therapy; HR, hazard ratio; CI, confidence interval; IM-IFNβ-1a, intramuscular interferon beta-1a; SC-IFNβ-1b, subcutaneous interferon beta-1b; SC-IFNβ-1a, subcutaneous interferon beta-1a; SUS, Brazilian Public Health System.

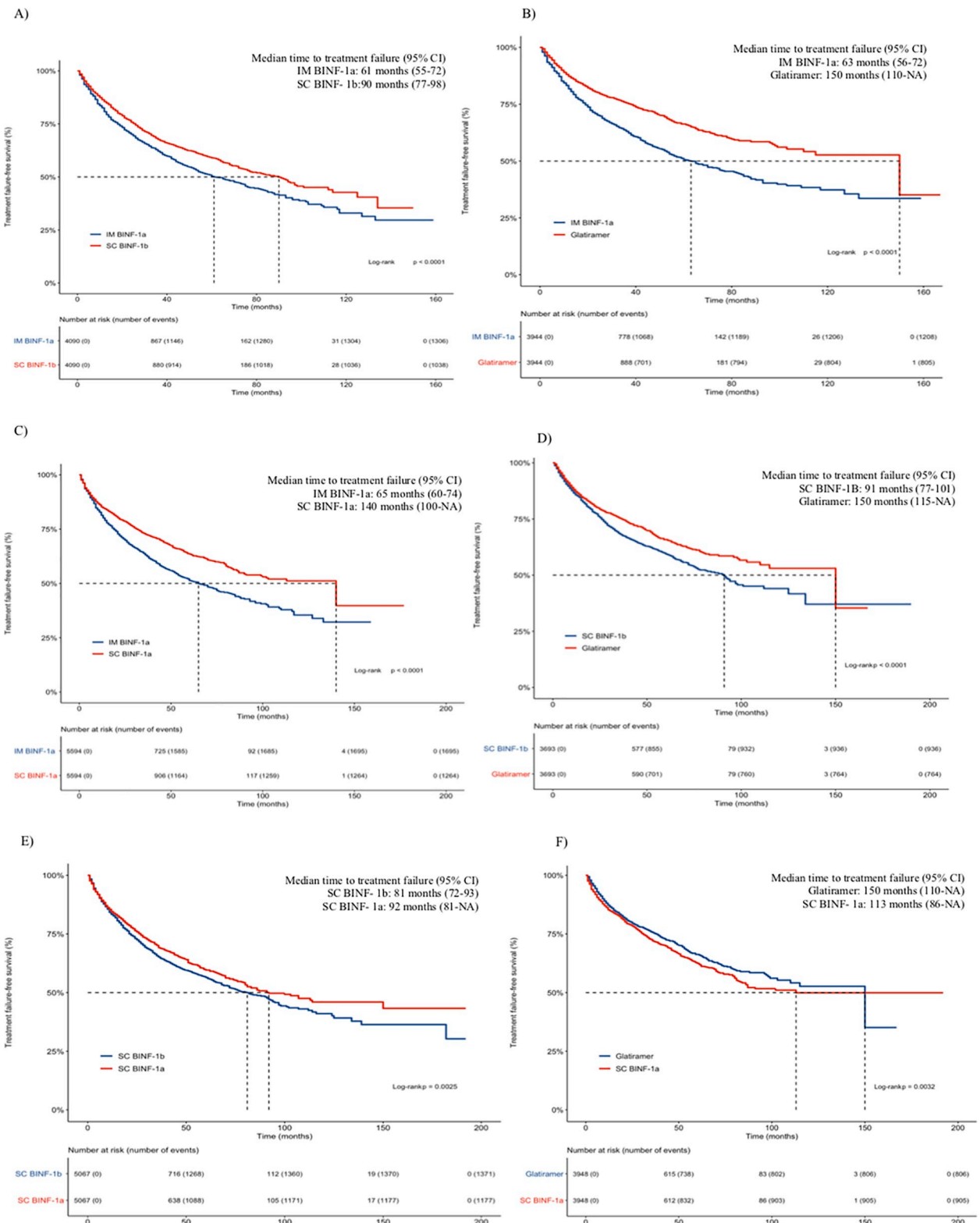

**Fig 3.** Kaplan–Meier matched comparisons for the time to treatment failure of IM βINF- 1a vs. SC βINF-1b (A), IM βINF- 1a vs. glatiramer (B), IM βINF- 1a vs. SC βINF-1a (C), SC βINF- 1b vs. glatiramer (D) SC βINF- 1b vs. SC βINF-1a (E) and Glatiramer vs. SC βINF-1a (F). Bonferroni adjustment for multiple comparisons. The significance level is p ≤ 0.008 (6 comparisons). **Abbreviations:** IM-IFNβ-1a, intramuscular interferon beta-1a; SC-IFNβ-1b, subcutaneous interferon beta-1b; SC-IFNβ-1a, subcutaneous interferon beta-1a.

**Table 4. Cox regression of DMT treatment failure for RRMS patients included in the cohort after pairing by propensity score, 2000 to 2015, Brazil.**

| Treatment groups | Crude HR (95% CI) | P-value[a] | Adjusted[b] HR (95% CI) | P-value[a] |
|---|---|---|---|---|
| IM βINF- 1a vs. SC βINF-1b | 1.28 (1.18–1.39) | <0.0001 | 1.31 (1.21–1.43) | <0.0001 |
| IM βINF- 1a vs. glatiramer | 1.60 (1.46–1.75) | <0.0001 | 1.65 (1.51–1.80) | <0.0001 |
| IM βINF- 1a vs. SC βINF-1a | 1.41 (1.31–1.51) | <0.0001 | 1.45 (1.35–1.56) | <0.0001 |
| SC βINF- 1b vs. glatiramer | 1.25 (1.14–1.38) | <0.0001 | 1.24 (1.13–1.37) | <0.001 |
| SC βINF- 1b vs. SC βINF-1a | 1.13 (1.04–1.22) | 0.01 | 1.11 (1.02–1.20) | ns |
| Glatiramer vs. SC βINF-1a | 0.87 (0.79–0.95) | 0.01 | 0.87 (0.79–0.96) | 0.01 |

[a] Bonferroni adjustment for multiple comparisons. The significance level is p ≤ 0.008 (6 comparisons).

[b] Adjustment by comorbidities after the starting DMT treatment.

**Abbreviations:** RRMS, relapsing-remitting multiple sclerosis; DMT, disease-modifying therapy; HR, hazard ratio; CI, confidence interval; IM-IFNβ-1a, intramuscular interferon beta-1a; SC-IFNβ-1b, subcutaneous interferon beta-1b; SC-IFNβ-1a, subcutaneous interferon beta-1a; SUS, Brazilian Public Health System, ns, non-significant.

was 50 (95% IC 46–53) months for IM βINF-1a group; 56 (95% IC 52–61) months for SC βINF-1b group; 97 (95% IC 87–113) months for glatiramer group; 69 (95% IC 65–73) months for SC βINF- 1a group and; the overall median time to treatment failure was 64 (95% IC 62–67) months (log-rank p-value = <0.0001).

In the same way, adjusted Cox regression analyses showed that IM βINF-1a had a higher risk of treatment failure at 28% (HR = 1.28, 95% CI 1.19–1.38in comparison to SC βINF-1b; 64% (HR = 1.64, 95% CI 1.52–1.79) in comparison to glatiramer and; 41% (HR = 1.41, 95% CI 1.32–1.51) in comparison to SC βINF-1a. Regarding to the sensitivity analysis restricted to exclusive SUS patients, we observed survival probabilities and HRs comparable to the main analysis. However, the median time to treatment failure were significantly lower when compared to patients who access SUS only to obtain medications (48 (95% IC 44–54) *versus* 82 (95% IC 72–109) months for IM βINF-1a group; 64 (95% IC 55–75) *versus* 97 (95% IC 86-NA) for SC βINF-1b group; 96 (95% IC 66-NA) *versus* 150 (95% IC 150-NA) for glatiramer group and; 69 (95% IC 59–78) *versus* 140 (95% IC 120-NA) for SC βINF- 1a group p<0,0001).

## Discussion

To evaluate the long-term effectiveness of first-line DMT treatment in Brazil, we conducted a deterministic-probabilistic linkage of SUS administrative databases and extracted a cohort of RRMS patients. This study highlights there is an important difference among first-line treatments in terms of treatment failure.

Overall, this study reports that women, exclusive SUS patients, presence of Elixhauser comorbidities and IM βINF-1a are associated with increased treatment failure survival time. Also, IM βINF-1a had a significantly higher risk of treatment failure than patients treated with other first-line DMT and both subcutaneous presentations of interferon-beta were inferior to glatiramer, independently of age, gender, comorbidities, region of residence, period of cohort entry and exclusive SUS patients in a Cox regression model. In the paired propensity score match analysis, although a few differences were found in terms of significance, the superiority of other DMTs over IM βINF-1a was evident.

Treatment groups differed in several clinical and demographic characteristics. Usually, the allocation to therapy is determined by MS treatment centers, individual physician choice, disability grade, course of illness, and age. In the study by Kalincik et. al (2014), the authors identified that patients treated with SC βINF-1a had more severe disabilities compared to the other

groups, whereas the patients receiving IM βINF-1a had fewer disabilities. Patients who started with glatiramer were older than those treated with SC or IM βINF-1a [27].

The provision of high-cost drugs is generally not included in the services provided by private insurance operators, as is the case with DMTs for the treatment of RRMS. Therefore, private insurance patients access SUS to obtain the drugs, but the registry of relapse or other medical care in SUS Information Systems for these patients is deficient [17, 18]. The data obtained in this cohort corroborate the literature of SUS coverage, as we identified 41.9% patients that are exclusively SUS users, that is, they do not have private insurance. It should be emphasized that this number may be underestimated, considering that we have adopted conservative methodology to estimate the number of these patients.

Regarding medicines, approximately 83% of patients included in the cohort began treatment with interferon beta, with SC βINF-1a being the most dispensed (35.6%), followed by IM βINF-1a (25.7%). Glatiramer represented 17.4% of the total cohort patients. Until 2017, interferons beta and glatiramer were the only first-line drugs provided by SUS and are in accordance with current clinical guidelines for MS [9, 28, 29]. This pattern of use could be observed in the cohorts of Kalincik et al. (2014), Evans et al. (2012), Halpern et al. (2011) and Reynolds et al. (2010) [20, 27, 30, 31]. The switch of medication was the event that most contributed to the composite failure outcome and is in accordance with the definition advocated by the Brazilian Clinical Guideline of MS, where the exchange between drugs is allowed through evidence of treatment failure [9].

Several evidences suggest that the presence of other comorbidities is common in individuals with MS and may have a negative impact on clinical outcomes and treatment. In this study, we identified that 10% of patients included had one or more associated comorbidities at the beginning of treatment, most of them being conditions related to MS. However, the prevalence and methods of collecting and identifying some comorbidities in MS remain underreported. In this study, we used the algorithm proposed by Quan et al. [21], which collects information from inpatient care. Also, it is known that the notification and registration of comorbidities to obtain medicines for the treatment of MS in SUS is an optional field, so we believe that this prevalence may be even higher [32–34].

Furthermore, other studies had evaluated the association between comorbidities and the time to first drug change and identified that comorbidities were significantly associated with increased risk of switching due to treatment intolerance [35, 36].

Concerning to the increased risk of treatment failure among women, findings from other studies corroborate the results of this study. It is known that MS affects more women than men. Besides, the progression and severity of the disease may differ from person to person, however, in women, in addition to pregnancy and breastfeeding, which require immediate discontinuation of treatment and consequently the risk of progression, factors such as menstruation and menopause can influence symptoms and progression of MS [31, 37–39].

In this cohort, the treatment failure stratified by the initial drug indicated significant differences between the first-line treatment options. Patients treated with glatiramer, followed by SC βINF-1a and SC βINF-1b had a significantly lower risk of treatment failure (40%, 30% and 22% respectively) than those treated with IM βINF-1a. In the treatment matched comparisons, the risk of treatment failure is even greater. It is observed that glatiramer acetate is superior to SC βINF-1b and SC βINF-1a, and these are statistically more effective than intramuscular interferon beta.

These results confirm and corroborate previous non-concurrent cohorts. Zhornistky *et al.* (2015) evaluated the time to treatment interruption due to intolerance or inefficacy of the initial treatment and showed that patients treated with interferons had a 29% risk of discontinuing treatment when compared to patients treated with glatiramer (HR = 1.29, 95% CI, 1.11–

1.50). In the study of Reynolds *et al.* (2010), Kaplan Meier analyses demonstrated that the initial drug switch or treatment discontinuation was significantly lower in patients treated with glatiramer [27, 31, 40]. Besides the real-world evidence observed in cohorts, the results of clinical trials and systematic reviews support the results found in this study [41–43].

On the other hand, similar studies with different results were also observed [44, 45]. Although the reason for this variability is not clear, it is assumed that sample size, clinical and demographic differences in baseline characteristics among the treatments compared may have influenced the outcome. In addition, since the cohort is not randomized, other differences that are not known between the groups may have existed.

In this non-concurrent cohort, the analysis used comparisons between the treatments with uniform inclusion criteria and the current clinical protocols for MS in Brazil. The population heterogeneity is important, as it reflects how the use of medicines is in the real world and their effectiveness thereafter. In an attempt to adjust the distributions between groups and the potential confounders that could influence the effectiveness results, pairing by propensity score was performed. This analysis reinforced the main findings of effectiveness between the treatments.

The main limitation of this study is that the data collected come from the administrative systems databases developed by SUS were not designed to evaluate and follow clinical results. Consequently, there is no record of patient-level clinical variables such as disability progression (EDSS scale), RMI results, disease duration, adverse event data, and causes that led to treatment switching. Besides, there is a possibility of incorrect data caused by sub-registers that may culminate in the sub or overestimation of the analyzes performed. It is likely, that many cases of relapses and other events have not been recorded in the SUS databases, as it is known that many patients use private services.

To overcome these limitations, we chose to use the treatment failure as composed outcome and persistence, capturing the switch of treatment that, as indicated in the clinical protocol, occurs in cases of relapses, intolerance or adverse events. Additionally, to minimize possible inconsistencies, after data collection, a sample inspection was performed with subsequent cleaning and standardization, constituting important steps to guarantee the quality of the collected data.

## Conclusions

This non-concurrent cohort containing more than twenty thousand patients with RRMS suggests that glatiramer acetate is associated with greater effectiveness compared to βIFN. When evaluating interferons beta, the results suggest that IM βIFN- 1a is the least effective in the treatment of RRMS.

Considering that the main objective of DMTs is to reduce the occurrence of relapses and, consequently, to reduce the progression of the disease, the inferiority of IM βIFN-1a in comparison to the other medications surpasses the convenience of one application per week. In addition, RRMS is a disease that has a large therapeutic arsenal available in the SUS and the recent updates of the Brazilian clinical guideline of Multiple Sclerosis count on new treatments, including oral medications. Therefore, considering the effectiveness of the treatment, the existence of therapeutic alternatives that present a better relation between cost, effectiveness and expected benefits for the patients is paramount for the good practices in the health system. The results of this research demonstrate the need to evaluate the real performance of DMTs and, based on these results, to encourage investment or disinvestment decisions in the health systems.

## Supporting information

**S1 Table. Characteristics of Brazilian patients with RRMS included in the study after matching by propensity score, 2000–2015.**
(DOCX)

## Author Contributions

**Conceptualization:** Kathiaja Miranda Souza, Augusto Afonso Guerra Júnior.

**Data curation:** Kathiaja Miranda Souza, Augusto Afonso Guerra Júnior.

**Formal analysis:** Kathiaja Miranda Souza, Isabela Maia Diniz, Augusto Afonso Guerra Júnior.

**Investigation:** Kathiaja Miranda Souza, Isabela Maia Diniz, Lívia Lovato Pires de Lemos, Augusto Afonso Guerra Júnior.

**Methodology:** Kathiaja Miranda Souza, Augusto Afonso Guerra Júnior.

**Project administration:** Augusto Afonso Guerra Júnior.

**Resources:** Kathiaja Miranda Souza.

**Software:** Augusto Afonso Guerra Júnior.

**Supervision:** Kathiaja Miranda Souza, Álvaro Nagib Atallah, Augusto Afonso Guerra Júnior.

**Validation:** Kathiaja Miranda Souza, Augusto Afonso Guerra Júnior.

**Visualization:** Kathiaja Miranda Souza, Augusto Afonso Guerra Júnior.

**Writing – original draft:** Kathiaja Miranda Souza, Isabela Maia Diniz, Lívia Lovato Pires de Lemos.

**Writing – review & editing:** Kathiaja Miranda Souza, Isabela Maia Diniz, Lívia Lovato Pires de Lemos, Nélio Gomes Ribeiro Junior, Isabella de Figueiredo Zuppo, Juliana Alvares Teodoro, Francisco de Assis Acurcio, Álvaro Nagib Atallah, Augusto Afonso Guerra Júnior.

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
