## [Decision Letter · Decision Letter 0]

23 Jun 2020

PONE-D-20-12343

Effectiveness of first-line treatment for relapsing-remitting multiple sclerosis in Brazil: A 16-year retrospective cohort study

PLOS ONE

Dear Dr. Souza,

Thank you for submitting your manuscript to PLOS ONE. After careful consideration, we feel that it has merit but does not fully meet PLOS ONE’s publication criteria as it currently stands. Therefore, we invite you to submit a revised version of the manuscript that addresses the points raised during the review process.

We look forward to receiving your revised manuscript.

Kind regards,

Sreeram V. Ramagopalan

Academic Editor

PLOS ONE

Journal Requirements:

2.We note that you have indicated that data from this study are available upon request. PLOS only allows data to be available upon request if there are legal or ethical restrictions on sharing data publicly. For information on unacceptable data access restrictions, please see http://journals.plos.org/plosone/s/data-availability#loc-unacceptable-data-access-restrictions.

3.Thank you for stating the following financial disclosure:

 'The funders had no role in study design, data collection and analysis, decision to publish, or preparation of the manuscript.'

Reviewers' comments:

Reviewer's Responses to Questions

**Comments to the Author**

1. Is the manuscript technically sound, and do the data support the conclusions?

Reviewer #1: Yes

2. Has the statistical analysis been performed appropriately and rigorously? 

Reviewer #1: Yes

3. Have the authors made all data underlying the findings in their manuscript fully available?

Reviewer #1: No

4. Is the manuscript presented in an intelligible fashion and written in standard English?

Reviewer #1: No

5. Review Comments to the Author

Reviewer #1: The authors have undertaken an impressive amount of work in linking together these databases, and conducting a large number of analyses. I was concerned about potential introduction of immortal time (see major comment 2), for which I’m suggesting some additional analyses (or, if I’ve misunderstood the design, clarifications). The paper also needs a significant amount of editing prior to publishing for readability, and it is very long at the moment. I would consider dropping one or more of the objectives and/or significantly shortening the introduction and discussion. I’m therefore recommending major revisions.

Please note that, based on my expertise, I’ve only been able to review the pharmacoepidemiological aspects of the paper (and not the clinical side).

Major comments

1.The paper contains the results from a large number of objectives (comorbidity development, factors associated with treatment failure, comparative effectiveness, persistence analyses) for a large number of drugs. Have the authors considered splitting the paper into two? Alternatively, I would recommend a significant amount of editing to shorten this paper to a standard length.

2.It seems that all patients were required to have at least 6 months of follow-up (row 206-207), however, it then seems patients were followed from the first dispensation. It is unclear to me whether this introduction of immortal time was intentional, and if yes, why the start of follow-up didn’t simply start from 6 months after the first dispensation. I would suggest that the authors evaluate whether this requirement of 6 months records of dispensation had an impact on the results, at least in a sensitivity analysis. Alternatively, if this was a requirement for 6 months of dispensation prior to the start of follow-up (as the first row of the results suggests) then this should be clarified.

Minor comments

1.Row 257. I’m not sure what the authors refer to when they state they performed a pairwise comparison of logistic regression analysis.

2.Table 4. The authors could consider scaling age to present it by 5 or 10 years, so that the effect estimates are easier to interpret.

3.Table 5. I’m not clear on the adjusted analyses in the propensity score matched cohorts, which is introduced in the results. I assume the additional comorbidities were included as time-updated variables, but this is not described in the methods and should be added.

4.I would recommend including the number under FU at each time-point per group in each of the KM plots

6. PLOS authors have the option to publish the peer review history of their article (what does this mean?). If published, this will include your full peer review and any attached files.

Reviewer #1: No

---

## [Author Response · Author response to Decision Letter 0]

14 Aug 2020

I have incorporated all of your suggestions into my revision. They were very helpful. Thank you.

---

## [Editor Report · Decision Letter 1]

18 Aug 2020

Effectiveness of first-line treatment for relapsing-remitting multiple sclerosis in Brazil: A 16-year non-concurrent cohort study

PONE-D-20-12343R1

Dear Dr. Souza,

We’re pleased to inform you that your manuscript has been judged scientifically suitable for publication and will be formally accepted for publication once it meets all outstanding technical requirements.

Kind regards,

Sreeram V. Ramagopalan

Academic Editor

PLOS ONE
---

## [Editor Report · Acceptance letter]

21 Aug 2020

PONE-D-20-12343R1 

Effectiveness of first-line treatment for relapsing-remitting multiple sclerosis in Brazil: A 16-year non-concurrent cohort study 

Dear Dr. Souza:

I'm pleased to inform you that your manuscript has been deemed suitable for publication in PLOS ONE. Congratulations! Your manuscript is now with our production department. 

Kind regards, 

on behalf of

Dr. Sreeram V. Ramagopalan 

Academic Editor

PLOS ONE